# The Management of Chronic Pain: Re-Centring Person-Centred Care

**DOI:** 10.3390/jcm12226957

**Published:** 2023-11-07

**Authors:** Kristy Themelis, Nicole K. Y. Tang

**Affiliations:** Department of Psychology, University of Warwick, Coventry CV4 7AL, UK

**Keywords:** mental defeat, sleep, digital health, chronic pain management, person-centred care, individual formulations, self, personalised pain management

## Abstract

The drive for a more person-centred approach in the broader field of clinical medicine is also gaining traction in chronic pain treatment. Despite current advances, a further departure from ‘business as usual’ is required to ensure that the care offered or received is not only effective but also considers personal values, goals, abilities, and day-to-day realities. Existing work typically focuses on explaining pain symptoms and the development of standardised interventions, at the risk of overlooking the broader consequences of pain in individuals’ lives and individual differences in pain responses. This review underscores the importance of considering additional factors, such as the influence of chronic pain on an individual’s sense of self. It explores innovative approaches to chronic pain management that have the potential to optimise effectiveness and offer person-centred care. Furthermore, it delves into research applying hybrid and individual formulations, along with self-monitoring technologies, to enhance pain assessment and the tailoring of management strategies. In conclusion, this review advocates for chronic pain management approaches that align with an individual’s priorities and realities while fostering their active involvement in self-monitoring and self-management.

## 1. Introduction

Chronic pain refers to pain that persists or recurs for more than three months [1]. It affects a significant number of individuals globally [2], with estimates reaching 1.5 billion people. It is a major contributor to years lost to disability and carries a substantial economic burden [2,3,4]. The toll of chronic pain is even more pronounced among children and adolescents, with disproportionally high costs [5]. In the United States alone, annual losses are projected to reach a staggering USD 19.5 billion [6]. Living with chronic pain can significantly impact a person’s emotional, physical, and social well-being. Research indicates that individuals who experience severe distress and disability due to chronic pain may also suffer from a compromised sense of self and identity [7,8,9]. Patients have expressed sentiments such as “I have lost my battle with pain to keep a hold on to my personal sense of self” and “I don’t like what I have become, and I feel negative towards myself and other people” [10] (p. 470). Accounts like these highlight the significant emotional and psychological impact that chronic pain can have on individuals, extending beyond the physical symptoms they experience [11,12]. 

Despite the significant personal impact of chronic pain, current interventions and strategies often focus on generic pain management approaches without necessarily addressing the personal meaning or psychological impact of pain on individuals. In a 2020 Cochrane review, 75 randomised controlled trials (RCTs) assessing psychological therapies for chronic pain management were examined for their clinical efficacy and safety in alleviating pain, reducing disability, and enhancing mood—these being the most commonly targeted and measured treatment outcomes in RCTs [13]. However, these conventional outcome measures may not fully capture the broad range of outcomes valued by individuals living with chronic pain [14]. It is worth noting that systematic reviews and meta-analyses, while informative, are constrained to extracting data solely from outcomes measured in RCTs, which inherently restricts the scope of available outcomes for analysis. Apart from pain reduction, improvements in pain interference, physical function, and emotional well-being, patients have consistently emphasised the significance of considering additional outcomes in both clinical trials and treatment. These include reductions in opioid use, alleviation of fatigue, improved sleep, and an enhanced sense of enjoyment of life [15,16,17]. The limited success of current clinical interventions highlights the need to better understand the lived experiences of individuals with chronic pain [18,19] and calls for more person-centred treatment evaluation [20,21]. It is important to acknowledge that certain clinical settings and healthcare professionals have already adopted a person-centred approach to address a range of health needs. However, there remains a broader challenge in ensuring that such patient-centred approaches are consistently integrated into multidisciplinary management protocols, making them a standard practice throughout the healthcare system, with a specific emphasis on specialised pain services [22,23]. 

This narrative review provides a concise overview of the existing body of research concerning psychological interventions for chronic pain. It underscores the importance of integrating patient-centred outcomes and explores the concept of the self within the context of chronic pain. The review also discusses the importance of person-centred care, examines the integration of digital technology, and explores individual formulation. The paper concludes with future directions, outlining key areas of focus and the next steps towards achieving person-centred care for individuals with chronic pain. 

## 2. Background on Psychological Interventions for Chronic Pain

Understanding the impact of pain on individuals requires acknowledging the influence of social and emotional factors, as proposed by Melzack and Wall’s (1965) gate control theory of pain [24]. Subsequent theoretical models have further emphasised the diverse ways in which people experience, interpret, and communicate pain [25]. In this narrative review, we centre our attention on the psychosocial aspects of chronic pain and its management. Here, we present a selection of the most frequently employed psychological interventions for chronic pain.

Psychological interventions for chronic pain primarily draw from cognitive and behavioural theories (CBT). One prominent psychological model is the fear-avoidance model [26,27,28,29], which proposes that when a person experiences pain, they may follow one of two paths, ultimately leading to either recovery or persistent pain [27]. The path to recovery is characterised by a low-threat appraisal of the pain, and a focus on engaging in activities that hold personal value. On the contrary, the path to persistent pain involves a heightened perception of pain as a threat. In this scenario, attempts to manage the pain are influenced by beliefs about pain and negative emotions, resulting in fear of pain and avoidance of activities. This disrupts daily life, amplifies negative emotions, and sets off a vicious cycle [27]. This model has gained significant influence and has sparked extensive research. It has also informed graded exposure-based treatments [30,31], where patients establish a hierarchy of fears and participate in behavioural experiments specifically designed to reduce fears and avoidance of valued activities [32]. This is achieved through graded exposure, which involves gradually and incrementally facing feared situations or activities according to a predetermined hierarchy that progresses from lower to highly anxiety-inducing situations. These treatments can also be delivered using virtual reality [33].

In terms of the social aspects of pain, the leading theory is the social communication model of pain [34], which emphasises the importance of the social context surrounding pain. It goes beyond just the individual experiencing pain and acknowledges the impact of caregiver perspectives, pain management strategies, knowledge, biases, and personal judgments. The dynamics between healthcare providers and patients, including factors like gender, power dynamics, and previous healthcare experiences, also influence pain experience [34]. Furthermore, the misdirected problem-solving model [35] assumes that, by the time pain becomes chronic, it is no longer directly associated with the original injury or tissue damage and becomes intractable by nature. In this context, the model suggests that attempting to solely address the pain through traditional problem-solving focused on pain reduction may not be effective because the underlying cause, such as the injury or tissue damage, may no longer be the primary driver of the ongoing pain experience. Instead, the model suggests that reframing the problem, often as the pursuit of finding better ways to live with the pain, results in a shift towards pursuing valued goals. This shift reduces the overwhelming dominance of pain-related concerns and reduces the persistent worry that often keeps individuals feeling ‘stuck’. Repeated attempts at problem-solving can inadvertently narrow the focus on the pain problem, trapping individuals in a cycle of rumination without an apparent solution [35].

Cognitive behavioural therapy is the most-delivered therapy for patients with chronic pain and often includes cognitive restructuring, problem solving, relaxation, activity pacing, and relapse prevention [36]. These approaches draw from the cognitive, behavioural, and social learning theories of psychological therapies [37,38] adapted to patients with chronic pain. Acceptance and Commitment Therapy (ACT) and mindfulness, often referred to as the “third wave”, have also gained prominence in pain management [39]. These approaches emphasise psychological flexibility, cognitive defusion, acceptance, moment-to-moment awareness, self-as-context, values orientation, and committed action [40,41]. The available evidence on ACT, as emphasised in a recent Cochrane review, is presently considered relatively limited and has prompted recommendations for a more rigorous evaluation [13]. Nonetheless, it has shown promise in various systematic reviews and meta-analyses [42,43,44]. The recent National Institute for Health and Clinical Excellence (NICE) guidelines from the UK have advised clinicians to consider ACT for chronic pain management [45]. This underscores a significant disparity and the need for a more comprehensive understanding of ACT’s effectiveness in comparison to control and other treatment approaches for managing chronic pain. 

In the field of chronic pain management, randomised controlled trials have been conducted extensively to determine the efficacy and safety of psychological interventions [46]. Meta-analyses have consistently demonstrated small-to-moderate improvements in reducing pain intensity and disability outcomes for various chronic pain conditions in both children and adults [13,47,48,49,50,51]. These improvements have been observed when comparing these interventions to “active control” conditions (such as education), “treatment as usual”, or “waiting list control” conditions, with the effects persisting for up to 12 months following therapy. However, the evidence base has shown limited changes in effect sizes over time [52,53,54], highlighting the need to further optimise psychological interventions and the necessity to explore novel psychological interventions that may hold the potential for more significant benefits. Additionally, there is growing recognition of the need to broaden the range of outcome measures in order to better capture the multifaceted impact of chronic pain interventions [55,56,57].

In summary, the field of chronic pain management has seen significant progress through psychological interventions, yet challenges persist in maximising their effectiveness. The diverse nature of pain experiences and the complex interplay of factors necessitate a broader exploration of novel therapies and a deeper understanding of the underlying mechanisms driving positive changes. Factors such as pain catastrophising [58,59], pain beliefs [60], and coping strategies [61] are commonly investigated, but consensus on the mechanisms responsible for positive changes in patients with chronic pain remains elusive. It is important to note that many of these therapies are often delivered in group settings and may not be necessarily tailored to individual needs. Therefore, ongoing research is vital for offering more tailored and effective approaches to alleviate the burden of chronic pain. 

## 3. The Self in Chronic Pain

In recent years, there has been an expansion and broadening of approaches to pain management. The focus has shifted from solely addressing pain to encompassing patients’ experiences and outcomes, which has become a required and essential component of treatment evaluation and audit processes [62]. This includes recognising the detrimental consequences of pain on the person’s sense of self, which has been a neglected area of research [8,40,63]. Here, we have chosen to focus on how pain affects an individual’s sense of self as an example in order to highlight the need to align interventions with the specific requirements and desired outcomes valued by patients with chronic pain. This highlights the potential for the development of novel treatments tailored to address these clinical needs. Other themes emerging from patient accounts, such as enjoyment of life, can also be explored in a similar manner [14].

A recent synthesis of 11 meta-ethnographies which systematically synthesises data from multiple studies, incorporating the experiences of thousands of individuals with chronic pain, has revealed the struggle to maintain a sense of self [10]. The psychological consequences of living with chronic pain have been described as an assault on [8], or a threat to, the person’s sense of self [64]. This observation applies across the lifespan, including children and adolescents, who express how pain has affected their aspirations and identity [65,66]. These qualitative accounts often adopt a “patient-focused” interpretative phenomenological approach (IPA), which allows for a detailed exploration of participants’ perspectives and lived experiences with chronic pain [67,68]. Traditional quantitative research methods have proven challenging in investigating the impact of pain on the self, given the lack of consensus on its definition [8,40]. However, while not providing definitive answers, these approaches offer valuable insights for understanding complex experiences [67] and informing clinical practice [11,69]. Therefore, there is a need for both quantitative and qualitative research methods to comprehensively understand and develop theories for testing the relationship between chronic pain and the self.

Building on these findings, one notable theory, the self-pain enmeshment theory [70], has prompted experimental investigations into the effects of chronic pain on the self. This theory suggests that the enmeshment (or blurring) of pain, illness, and self can bias information processing and exacerbate distress in people with chronic pain. For someone experiencing enmeshment due to their pain, this might mean that their experience of pain becomes so entangled with who they are that it influences how they define themselves, intrudes upon various aspects of their life, and poses a significant threat to their sense of self. In this context, self-pain enmeshment is often operationalised to measure a person’s identity, encompassing their perception of who they are and their potential future self [64,70,71]. This has been measured through empirical studies using hierarchical regression analysis to examine the relationship between self-report scales and self-described aspects of individuals’ selves [71]. Participants in these studies were asked to generate up to 10 personal descriptors for their current self, hoped-for self, and feared-for self, and then evaluate whether these aspects remained possible or were contingent on the presence of pain. The findings of these investigations have provided evidence supporting a positive association between an individual’s desired self-image and the magnitude of depression [71]. The underlying theory suggests that as individuals become more enmeshed with their pain experience, the poorer the resulting psychological adjustment. This hypothesis finds support in quantitative evidence indicating a positive association between the extent pain- and self-schemata, as measured by the Implicit Association Test [72], and the levels of pain suffering, anxiety, and helplessness experienced by individuals [73]. 

Another line of research focuses on mental defeat, a concept applied to the study of chronic pain and based on the literature on post-traumatic stress disorder (PTSD) and depression [74,75,76,77,78]. Living with persistent and debilitating pain is seen as a repeated trigger of mental defeat, leading to negative self-appraisals [79] and encroachment on autonomy and personal integrity [8,80,81]. In empirical and prospective studies, the use of the Pain Self Perception Scale [79] has identified mental defeat as a significant predictor of various outcomes among patients with chronic pain. This includes pain interference, depression, psychological disability [80], and suicide risk or intent [82,83] within this population [84]. This aligns with established theories of suicide featuring defeat as a key cognitive factor driving suicidal thoughts and behaviour, such as the cry of pain model [85] and the schematic appraisal model [86].

In summary, shifts in the focus of pain management have underscored the significance of patient experience, including the impact of pain on their sense of self, which is an area requiring further empirical investigation. Concepts like self-pain enmeshment and mental defeat provide valuable insights into the effects of pain on individuals, offering potential avenues for further research and the innovative development of treatments that can be incorporated into existing psychological approaches for pain management. 

## 4. Person-Centred Care

As per the World Health Organization (WHO), person-centred care refers to healthcare approaches and methods that consider the individual as a unified entity with multifaceted needs and objectives which stem from their unique social determinants of health [87]. A person-centred approach to the assessment and management of chronic pain emphasises the importance of personalised care, where treatments are tailored to meet the individual’s needs [88,89,90]. This should not be confused with personalised medicine or precision medicine, which often relies on genetic information to guide decisions related to disease prevention, diagnosis, and treatment [91]. 

Efforts are underway to establish a consensus among patients, researchers, and clinicians regarding outcome measures for the psychosocial treatment of chronic pain. This also extends to the selection of response scales that align with the unique needs and perspectives of the individuals whose outcomes are being measured [92,93]. Initiatives such as the Initiative on Methods, Measurement, and Pain Assessment in Clinical Trials (IMMPACT) [94] aimed to recommend core outcome measures and domains for clinical trials but did not initially include the perspectives of individuals living with pain [93]. Recognising the importance of these perspectives, subsequent efforts have incorporated them [55,56,95], going beyond what has typically been assessed in clinical pain interventions. These include measures of fatigue, sleep, home and family care, engagement in social and leisure activities, interpersonal relationships, and sexual activities, in addition to the core outcome domains previously identified, such as pain relief and enhancements in both physical and emotional well-being [55,94]. These outcome measures also reflect the pervasive nature of chronic pain and extend beyond disease-specific outcomes, recognising the interconnectedness of various aspects of an individual’s well-being when living with chronic pain. 

The Department of Health in the United Kingdom champions a hybrid treatment approach that relies on evidence-based decision-making between healthcare providers and patients, considering the preferences of everyone involved [88]. The success of this approach hinges on broadening outcomes to incorporate the perspectives of patients and include areas that hold personal significance to them. This necessitates a departure from a primary focus on pain management and a move towards the diverse needs of individuals coping with pain. Numerous experts have endorsed a hybrid treatment approach that surpasses pain management alone, aiming to tackle a range of factors or comorbidities that influence both quality of life and daily functioning [18,96,97,98,99,100]. A hybrid approach offers a more comprehensive perspective on addressing the distress and disability experienced by individuals with chronic pain. It allows for a personalised approach by aligning treatment needs, rather than depending on predefined and standardised methods [96].

An illustrative example comes from a growing body of evidence supporting the effectiveness of hybrid treatments targeting sleep quality in patients with comorbid insomnia and chronic pain [96,101]. Sleep disturbances and chronic pain often co-exist and influence each other bidirectionally [102,103,104,105,106,107,108,109]. Research suggests that a significant proportion of individuals with chronic pain experience sleep disturbances, with prevalence rates reported as high as 75% [110]. Cognitive-behavioural approaches that target sleep disturbances in chronic pain, such as cognitive behavioural therapy for insomnia (CBT-I), have demonstrated efficacy in improving sleep symptoms and have small-to-medium effect sizes on pain outcomes [111,112,113]. Hybrid approaches that combine components of CBT-I and cognitive behavioural therapy for pain (CBT-P) are known as CBT-IP and include objectives related to sleep hygiene, sleep quality, and sleep patterns, as well as objectives aimed at addressing the thoughts, emotions, and behaviours associated with pain. Research has shown that these hybrid treatments hold promise in improving sleep and function [111,112,113]. A secondary analysis of a large randomised controlled trial involving older adults with comorbid osteoarthritis and insomnia revealed that CBT-IP led to significant improvements in pain compared to CBT-P alone, particularly in patients with higher levels of insomnia and pain severity at baseline [114]. However, the effects of CBT-IP on pain compared to CBT-I alone are mixed [111]. Expanding pain interventions to target sleep problems can bring several potential benefits. In addition to the potential cost reduction compared to implementing multiple separate treatments, it can make patients feel heard, enhance their understanding of the interplay between sleep and pain, and boost their confidence by achieving improvements in sleep, which may, in turn, positively impact other areas of their lives [101]. Focus group discussions with patients echo the importance of tailoring interventions within a broader hybrid CBT framework that caters to individual needs [115]. Similar hybrid approaches, combining interventions to address two or more comorbid conditions such as sleep and physical activity, are being developed using digitally delivered CBT [115]. 

Some interventions employ prognostic screening methods to stratify patients into treatment pathways based on prognostic factors or clinical needs. For instance, the STarT Back Trial [116] assigned patients with back pain to three risk-defined groups (low-, medium-, or high-risk) and provided specific treatment packages tailored to each group’s clinical needs. The use of prognostic screening and matched treatment pathways resulted in short-term health improvements, increased patient satisfaction, and reduced health care costs compared to best current non-stratified care for back pain patients [116]. However, the long-term benefits in terms of disability have not yet been demonstrated [116] and the predictive value of the stratification tool for pain-related outcomes was limited to disability [117]. Overall, emerging interventions aim to broaden and diversify the focus on pain management by simultaneously targeting pain, insomnia, and other comorbidities. The ability to adjust treatment based on individual needs offers promising avenues compared to standard pain management programs. 

## 5. Use of Digital Health Technology

The use of digital health technology has seen a significant increase in the field of chronic pain management [47,49]. Remotely delivered interventions through online platforms, e-health, or m-health [47,48,51,118,119] have demonstrated beneficial effects similar to face-to-face interventions [49], but questions remain about how they compare in regards to active control, intervention longevity, and potential harm [120,121]. Another trend is the use of autonomous interventions, including virtual reality or telephone voice-automated interventions [122]. The COVID-19 pandemic has further accelerated the adoption of online healthcare delivery, as many pain clinics temporarily closed during the pandemic [120]. Three key factors are driving the need for remotely delivered, e-health interventions: limited accessibility to specialist pain clinics located in urban locations; the absence of spontaneous recovery for individuals with chronic pain while waiting for treatment; and travel restrictions imposed during the pandemic. Digital technology can help to overcome some of the barriers associated with face-to-face interventions and provide alternative options for individuals living with pain who cannot access in-person treatment [123]. 

The integration of digital technology in healthcare has also allowed patients to have more access to their healthcare data. Access to healthcare data, web-based interventions, and self-monitoring tools are considered important for empowering individuals to take control of their health [124]. A meta-analysis of studies on telehealth interventions for chronic musculoskeletal pain found that flexibility in accessing interventions and empowering patients to self-manage their conditions were significant predictors of participation [121]. 

The integration of digital technology in research presents unique opportunities to combine self-report measures, psychological assessments, behaviour tracking, and the collection of environmental data in real-life settings over extended periods. Wearable tracking devices, such as those used for ecological momentary assessment (EMA), have gained popularity and enable the continuous monitoring of health status, including variables like physical activity, sleep, and physiological measures such as blood pressure, pulse, and temperature [125]. These measurements provide real-time assessment and tailored feedback to evaluate interventions, particularly in sleep treatment where portable devices have allowed monitoring outside the laboratory settings for extended periods [126]. These advancements provide valuable insights into the interplay between pain and sleep across the lifespan [112,127,128,129,130].

However, concerns persist regarding whether personalised treatment approaches using digital health technology are perceived as personal by patients. Qualitative research indicates that telehealth interventions may be viewed as impersonal or unengaging when there is a lack of connection with the clinician, especially in online interventions without physical presence or hands-on solutions [121]. Pain management apps available on the IOS and Android marketplace vary in terms of content and quality, with only a few incorporating interactive components or being explicitly based on psychological principles such as CBT, ACT, or mindfulness [121]. 

The rise of internet-delivered psychological therapies and the increasing use of wearable technology present opportunities to capture a wide range of relevant outcomes in clinical practice and trials [131]. Combining self-report measures with data collected through EMA can provide deeper insights into the interplay between various psychological and physiological factors, highlighting temporal associations and patterns. Additionally, web-based interactive platforms can offer modular and individualised treatment approaches, granting patients greater autonomy in choosing treatments at their convenience. The abovementioned developments are expected to lay the foundation for novel technological solutions in clinical and natural contexts. However, issues related to quality control, effectiveness, scientific validity, and personalisation need to be addressed before wider implementation [120,121]. 

Individual formulation plays a pivotal role in tailoring these hybrid approaches to address the unique needs of each chronic pain patient. This approach builds on the idea of shared decision-making of treatment and management options and makes it a collaborative endeavour [132]. By understanding an individual’s specific factors, needs, and goals, healthcare professionals can create personalised interventions that encompass not only pain management but also broader aspects of their well-being, including sleep, physical activity, and social relationships. This individualised approach aligns with the wider principles of person-centred care, where treatments are finely tuned to the preferences and needs of each patient. Despite the seeming dichotomy between personalisation and standardisation, insights from other sectors suggest a balance can be achieved [133]. This balance involves providing personalised care, enabled by IT and digital interventions, that is delivered within the structure of standardised processes [134,135], thereby targeting meaningful improvements in both clinical and non-clinical outcomes whilst building on the efficacy and predictability of existing processes and clinical pathways with the flexibility to cater to individual patient needs and goals. 

## 6. Summary and Future Directions

In summary, psychological interventions for chronic pain have been extensively developed and refined. However, the focus of these interventions remains largely on pain symptom management and less on outcomes that are significant to individuals managing pain [14]. The field faces challenges regarding the magnitude of treatment effects [136], necessitating further innovative advancements. We believe several key changes can drive progress in the field. 

In the impactful words of Lyman (2021), a doctor who, early in his career, realised the detrimental consequences of misunderstanding pain and denying pain relief to others: “Pain isn’t found in the body, but it also isn’t just found in the mind: pain is in the person. To treat pain, we need to treat the whole human. Recovery is changing the meaning of pain; it is about recovering identity and personhood” [137] (p. 201).

Firstly, the management of and interventions for chronic pain should revolve around the values and preferences of individuals living with persistent pain, and outcomes should be selected accordingly in collaboration with them, rather than solely by clinicians and researchers [14,17]. Through individual formulation, research should build upon existing psychological therapies and adopt person-centred approaches to identify what matters most to each individual, expanding the range of outcomes typically measured. This involves establishing an evidence base to better understand how pain affects a person’s identity and sense of self and what can be done to help people rebuild and renew their identities, or even forge new ones. 

Efforts should be made to ensure that outcomes align with the individual’s priorities, abilities, and goals and that these are consistently measured across studies to ensure comparability. A hybrid approach that incorporates relevant elements from existing CBT interventions could broaden the focus of treatments to include goals associated with pain, such as improving sleep and increasing physical activity. Embracing a transdiagnostic management approach that targets common comorbidities could empower patients to have more control over their treatment plan and shift the focus away from mostly unattainable goals like complete pain relief [35,96,138]. Recommendations for future research include conducting RCTs to compare different modes of treatment delivery, treatment content, control groups, and long-term follow-up, all with a strong focus on personalisation [45].

Advancements in technologies and online tools provide opportunities for person-centred care, enabling patients to self-monitor and self-manage their pain with or without clinician guidance. However, further empirical work is needed to critically evaluate the effectiveness of remotely delivered management services and to tailor them to the specific needs of individuals with chronic pain. By focusing on what is important to the person living with pain and incorporating dedicated content addressing frequently reported comorbidities, the outlined steps have the potential to propel the field forward. These steps aim to address aspects that significantly impact quality of life but are currently overlooked or inadequately targeted in healthcare and pain management programs. 

## Data Availability

Not applicable.

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
