# Peer review of "The Management of Chronic Pain: Re-Centring Person-Centred Care"

_jcm, 2023, doi:10.3390/jcm12226957_

Round 1

Reviewer 1 Report

Comments and Suggestions for Authors

I appreciate the opportunity to review the submitted paper. I think the topic is important and worthy of a review. I will also compliment the authors on their writing, which was clear and efficient. There are some more critical comments below which I hope the authors will take in a spirit of constructive feedback. 

1.       Thinking about the title first, I expected that “imagination” would play some part in the narrative but I did not see that it did, so this reference seemed a bit disconnected.  Another notion that struck me is that while research trials are often constrained by outcomes that are not set by the individual recipients, actual practice is often more individually focused, and much of current practice is person-centered or personalized, at least to a degree. I see this distinction between research and practice as unaddressed in the paper.

2.       The authors don’t seem to provide a detailed definition of person-centered care. I wonder if they need to do this. They seem to implicitly reduce it to patient chosen outcomes, and treatments organized according to patient problem lists.  Is this really all it is?

3.       I found the arbitrary moving back and forth between studies of adults and young people distracting and could not understand the motivation for doing this.

4.       The authors have a section on the background of psychological interventions for chronic pain. In this section they selectively feature mainly different models and say relatively little about treatments. In fact, some of the models presented have not been translated into treatment methods at all, and I do not see their relevance to the stated topic of the section.

5.       Third wave therapies do not emphasize cognitive “diffusion.” The word is defusion.

6.       There are more than 30 RCTs of ACT and most systematic reviews and meta-analyses, including at least four in number, suggest that it is beneficial for people with pain. It is only the Cochrane reviews, featuring a peculiarly small selected set of RCTs, that say otherwise. Numerous studies show that convention CBT and ACT are not different in efficacy. While some researchers are waiting for more rigorous evaluations, those who are well-informed and up to date are not.

7.       Most of the studies of the pain-enmeshment theory are quite old, with perhaps the key review on the related concepts appearing 22 years ago. Has this theory yet yielded any treatment developments?  If not, will it ever?

8.        It seems to me a more critical and practical perspective is required in relation to the concepts and evidence around self and chronic pain. What is the quality of evidence around mental defeat for example? Are there prospective, experimental, or mediation studies of it. Does it lend itself to the design and development of new treatment methods that are distinct from those we already have?

9.       The presentation of “hybrid” treatment approaches seems to include adding two treatments together, unless I am misunderstanding. Will this strike readers as highly personal or individual?

10.   There seems to be an insinuation that pain management is just focused on pain. I believe this is not the case either in research or in clinical practice of psychological approaches, most of the time.

Author Response

Summary

Thank you very much for taking the time to review this manuscript. Please find the detailed responses below and the corresponding revisions highlighted/in track changes in the re-submitted files.

Comments 1:  Thinking about the title first, I expected that “imagination” would play some part in the narrative but I did not see that it did, so this reference seemed a bit disconnected. 

Another notion that struck me is that while research trials are often constrained by outcomes that are not set by the individual recipients, actual practice is often more individually focused, and much of current practice is person-centered or personalized, at least to a degree. I see this distinction between research and practice as unaddressed in the paper.

Response 1: Thank you for pointing out the wording in the title. We have revised the title without the “Imagining” part to better reflect the content of the paper and to acknowledge that this is not a new movement. “The Management of Chronic Pain: Re-centring Person-centred Care”.

We also appreciate the observation regarding the disconnect between research and clinical practice, particularly in the context of person-centred care. We agree with the comment regarding the divide. However, we do believe there is an existing gap for research findings and management approaches to become standard practice. We have therefore clarified the text to emphasise that patient outcomes refer both to clinical trials and treatment.

We have made the following changes in the manuscript (Page 3, line 66, and lines 70-76), with the modifications in red below:

Apart from pain reduction, improvements in pain interference, physical function, and emotional well-being, patients have consistently emphasised the significance of considering additional outcomes in both clinical trials and treatment.  

These include reductions in opioid use, alleviation of fatigue, improved sleep, and an enhanced sense of enjoyment of life [15–17]. The limited success of current clinical interventions highlights the need to better understand the lived experiences of individuals with chronic pain [18,19] and calls for more person-centred treatment evaluation [20,21]. It is important to acknowledge that certain clinical settings and healthcare professionals have already adopted a person-centred approach to address a range of health needs. However, there remains a broader challenge in ensuring that such patient-centred approaches are consistently integrated into multidisciplinary management protocols, making them a standard practice throughout the healthcare system, with a specific emphasis on specialised pain services (National Pain Audit, 2010-2012; Price et al., 2019)

Comments 2:  The authors don’t seem to provide a detailed definition of person-centered care. I wonder if they need to do this. They seem to implicitly reduce it to patient chosen outcomes, and treatments organized according to patient problem lists.  Is this really all it is?

Response 2: Agree. We have added the following definition to the section on Person-Centred care (page 5, lines 484-486):

As per the World Health Organization (WHO), Person-centred care refers to healthcare approaches and methods that consider the individual as a unified entity with multifaceted needs and objectives, which stem from their unique social determinants of health (WHO, 2015).

Comments 3: I found the arbitrary moving back and forth between studies of adults and young people distracting and could not understand the motivation for doing this.

Response 3: We appreciate the reviewer’s input and have removed certain elements, to enhance the clarity of the manuscript. We do believe in the importance of highlighting chronic pain experiences across the lifespan to understand the diverse challenges individuals face. We believe that the revised section now effectively conveys this perspective while addressing the concerns raised by the author.

Deleted section (Page 3)

The Fear Avoidance model described above has undergone variations and reformulations [25,32–34], which continues to inspire further research. One notable variation focuses on children with pain and their parents and families [35], emphasising the significance of communication between parent and child regarding pain and functioning. It highlights the impact of parental protective behaviours, which have been associated with child disability [36]. Additionally, a recent review found that factors such as parent pain catastrophising, anxiety, depression, and stress related to parenting a child with chronic pain were associated with increased child pain intensity and disability [37]. This underscores the enmeshed bidirectional relationship between children and their parents and emphasises the vital role of effective communication more generally.

Comments 4:   The authors have a section on the background of psychological interventions for chronic pain. In this section they selectively feature mainly different models and say relatively little about treatments. In fact, some of the models presented have not been translated into treatment methods at all, and I do not see their relevance to the stated topic of the section.

Response 4:

The section on the background of psychological interventions for chronic pain was included to provide a brief overview of foundational models and concepts. We acknowledge that not all the models presented have direct treatment applications, but they have played a significant role in shaping our understanding of chronic pain and it serves as a crucial foundation for exploring innovative approaches to chronic pain management.

We have incorporated a disclaimer to the section to underscore our primary focus (Background on psychological interventions for chronic pain, Page 2, lines 89-92).

Understanding the impact of pain on individuals requires acknowledging the influence of social and emotional factors, as proposed by Melzack and Wall’s (1965) gate control theory of pain [24]. Subsequent theoretical models have further emphasised the diverse ways in which people experience, interpret, and communicate pain [25]. In this narrative review, we centre our attention on the psychosocial aspects of chronic pain and its management. Here, we present a selection of the most frequently employed psychological interventions for chronic pain.

Comments 5:  Third wave therapies do not emphasize cognitive “diffusion.” The word is defusion.

Response 5: Thank you for spotting the error. We have corrected this to “Defusion” (Page 3, line 136)

Comments 6:  There are more than 30 RCTs of ACT and most systematic reviews and meta-analyses, including at least four in number, suggest that it is beneficial for people with pain. It is only the Cochrane reviews, featuring a peculiarly small selected set of RCTs, that say otherwise. Numerous studies show that convention CBT and ACT are not different in efficacy. While some researchers are waiting for more rigorous evaluations, those who are well-informed and up to date are not.

Response 6:  We appreciate the perspective on the comparative efficacy of CBT and ACT and noted the point about the Cochrane reviews featuring a specific set of RCTs. We are aware of the non-Cochrane systematic reviews and meta-analyses that the reviewer is referring to, and we have revised the current section to ensure it accurately represents the current state of research and the diverse viewpoints in the field by revising the section on page 3.

The revised text reads as follows (Page 3, lines 189-196).

The available evidence on ACT, as emphasised in a recent Cochrane review, is presently considered relatively limited and has prompted recommendations for a more rigorous evaluation [13]. Nonetheless, it has shown promise in various systematic reviews and meta-analyses [44–46]. The recent National Institute for Health and Clinical Excellence (NICE) guidelines from the UK has advised clinicians to consider ACT for chronic pain Management [47] .This underscores a significant disparity and the need for a more comprehensive understanding of ACT’s effectiveness in comparison to control and other treatment approaches for managing chronic pain.

Comments 7:  Most of the studies of the pain-enmeshment theory are quite old, with perhaps the key review on the related concepts appearing 22 years ago. Has this theory yet yielded any treatment developments?  If not, will it ever?

Comments 8:  It seems to me a more critical and practical perspective is required in relation to the concepts and evidence around self and chronic pain. What is the quality of evidence around mental defeat for example? Are there prospective, experimental, or mediation studies of it. Does it lend itself to the design and development of new treatment methods that are distinct from those we already have?

Response 7&8

While it's true that some of the foundational research in this area dates back a number of years, it provides valuable insights into the impact of chronic pain on individuals. Although these theories may not have directly led to treatment developments, they offer potential for future research and the development of treatments that can be integrated into existing psychological approaches for pain management. While our manuscript acknowledges the existing evidence base, we also stress the importance of further research. We firmly believe that the current evidence provides a robust foundation for understanding these concepts, but we remain committed to the ongoing need for exploration and refinement. We also highlight the potential significance of these concepts in innovative interventions for chronic pain.

Moreover, we wish to draw attention to a sentence in the summary and future directions section (Page 752-755) (“ This involves establishing an evidence base to better understand how pain affects a person’s identity and sense of self and what can be done to help people rebuild/renew their identity or even forge new ones”), which emphasizes the importance of establishing an evidence base to better comprehend how pain affects a person's identity and sense of self.

We have also added a qualifying sentence (Page 5, lines 479-484) to underscore the need for further investigation:

In summary, shifts in the focus of pain management have underscored the significance of patient experience, including the impact of pain on their sense of self, which is an area requiring further empirical investigation. Concepts like pain-self enmeshment and mental defeat provide valuable insights into the effects of pain on individuals, offering potential avenues for further research and the innovative development of treatments that can be incorporated into existing psychological approaches for pain management.

Comments 9:  The presentation of “hybrid” treatment approaches seems to include adding two treatments together, unless I am misunderstanding. Will this strike readers as highly personal or individual?

Response 9:

We have made the following changes in the manuscript to clarify this point (Page 6, lines 567-571), Line 573, lines 581-584, lines 590-591 and lines 596-597), with the modifications highlighted in bold:

The Department of Health in the United Kingdom champions a hybrid treatment approach that relies on evidence-based decision-making between healthcare providers and patients considering the preferences of everyone involved [88]. The success of this approach hinges on broadening outcomes to incorporate the perspectives of patients and includes areas that hold personal significance to them. This necessitates a departure from a primary focus on pain management and a move towards the diverse needs of individuals coping with pain. Numerous experts have endorsed a hybrid treatment approach that surpasses pain management alone, aiming to tackle a range of factors or co-morbidities that influence both quality of life and daily functioning [18,95–99]. A hybrid approach offers a more comprehensive perspective on addressing the distress and disability experienced by individuals with chronic pain. It allows for a personalised approach by aligning treatment needs, rather than depending on predefined and standardised methods [104].

An illustrative example comes from a growing body of evidence supporting the effectiveness of hybrid treatments targeting sleep quality in patients with comorbid insomnia and chronic pain. Sleep disturbances and chronic pain often co-exist and influence each other bidirectionally [100–107]. Research suggests that a significant proportion of individuals with chronic pain experience sleep disturbances, with prevalence rates reported as high as 75% [108]. Cognitive-behavioural approaches that target sleep disturbances in chronic pain such as Cognitive Behavioural Therapy for Insomnia (CBT-I) have demonstrated efficacy in improving sleep symptoms and have small to medium effect sizes on pain outcomes [109–111]. Hybrid approaches that combine components of CBT-I and Cognitive Behavioural Therapy for Pain (CBT-P), known as CBT-IP include objectives related to sleep hygiene, sleep quality, and sleep patterns as well as objectives aimed at addressing the thoughts, emotions, and behaviours associated with pain. Research has shown that these hybrid treatments hold promise in improving sleep and function [109–111]. A secondary analysis of a large randomised controlled trial involving older adults with comorbid osteoarthritis and insomnia revealed that CBT-IP led to significant improvements in pain compared to CBT-P alone, particularly in patients with higher levels of insomnia and pain severity at baseline [112]. However, the effects of CBT-IP on pain compared to CBT-I alone are mixed [109]. Expanding pain interventions to target sleep problems can bring several potential benefits. In addition to the potential cost reduction compared to implementing multiple separate treatments, it can make patients feel heard, enhance their understanding of the interplay between sleep and pain, and boost their confidence by achieving improvements in sleep, which may, in turn, positively impact other areas of their lives [113]. Focus group discussions with patients echo the importance of tailoring interventions within a broader hybrid CBT framework that caters to individual needs [114]. Similar hybrid approaches, combining interventions to address two or more co-morbid conditions such as sleep and physical activity, are being developed using digitally delivered CBT [114].

Comments 10:   There seems to be an insinuation that pain management is just focused on pain. I believe this is not the case either in research or in clinical practice of psychological approaches, most of the time.

Response 10: We understand the viewpoint that pain management, especially when utilising psychological approaches, often extends beyond solely addressing pain. Our intention was not to insinuate otherwise, but rather to emphasise the significance of considering additional factors and dimensions in pain management. Your comment underscores to importance of formally recognising these practices, and in particular for psychological interventions.

We have made the following changes in the manuscript (Page 2, lines 70-74) with the modifications highlighted in bold.

The limited success of current clinical interventions highlights the need to better understand the lived experiences of individuals with chronic pain [18,19] and calls for more person-centred treatment evaluation [20,21]. It is important to acknowledge that certain clinical settings and healthcare professionals have already adopted a person-centred approach to address a range of health needs. However, there remains a broader challenge in ensuring that such patient-centred approaches are consistently integrated into multidisciplinary management protocols, making them a standard practice throughout the healthcare system, with a specific emphasis on specialised pain services [22,23].

Reviewer 2 Report

Comments and Suggestions for Authors

This manuscript introduces and comprehensively discusses the importance of considering additional factors, such as the influence of chronic pain on an individual’s sense of self.

The topic is original and relevant to the field. There is limited information on this topic in the literature.

This article makes clear and advocates for chronic pain management approaches that align with an individual's priorities and realities while fostering active involvement in self-monitoring and self-management.

There are no further improvements regarding the methodology.

The conclusions are consistent with the evidence and arguments presented as well as summarize the main point of this article. 

References are up-to-date and appropriate

Minor revision

1) I would like a brief discussion on pain rating scales such as VAS, and NRS.

2) According to the literature, NRS is usually chosen because compared to other pain intensity scales it is preferable by patients, as well as in comparison to other pain scales (such as the Visual Analogue Scale, VAS)

Add this information and consider citing:

https://pubmed.ncbi.nlm.nih.gov/33155461/

https://pubmed.ncbi.nlm.nih.gov/33424089/

Author Response

1. Summary

Thank you very much for taking the time to review this manuscript. We appreciate your positive comments regarding the manuscript's originality and relevance in the field, particularly given the limited existing literature on this topic. Please find the detailed responses below and the corresponding revisions/corrections highlighted/in track changes in the re-submitted files.

Comments 1 & 2:

- I would like a brief discussion on pain rating scales such as VAS, and NRS.

- According to the literature, NRS is usually chosen because compared to other pain intensity scales it is preferable by patients, as well as in comparison to other pain scales (such as the Visual Analogue Scale, VAS)

Add this information and consider citing:

https://pubmed.ncbi.nlm.nih.gov/33155461/

https://pubmed.ncbi.nlm.nih.gov/33424089/

Response 1 & 2: Thank you for highlighting the importance of response scale selection. Whilst a detailed discussion of preferences in pain rating scales is beyond the scope of our paper, we have incorporated a section that underscores the importance of selecting response scales more generally. We have opted to include references to review papers that provide a broader and more comprehensive perspective on this matter.

Added section (Page 5, lines 489-491):

This also extends to the selection of response scales that align with the unique needs and perspectives of the individuals whose outcomes are being measured (Safikhani et al. 2018; Dworkin et al. 2008).